# Weaning Performance of Beef Cattle Calves Based on Concentrate Intake

**DOI:** 10.3390/ani10010018

**Published:** 2019-12-20

**Authors:** Chong Wang, Dongping Li, Jinyong Yang, Yuefeng Xia, Yan Tu, Robin White, Hui Gao, Qiyu Diao, Huiling Mao

**Affiliations:** 1College of Animal Science and Technology College of Veterinary Medicine, Zhejiang A & F University, Lin’an, Hangzhou 311300, China; wangcong992@163.com (C.W.); lidongpingv0621@163.com (D.L.); xyf_azrael@163.com (Y.X.); 2Key Laboratory of Applied Technology on Green-Eco-Healthy Animal Husbandry of Zhejiang Province, Hangzhou 311300, China; 3Zhejiang Provincial Engineering Laboratory for Animal Health Inspection and Internet Technology, Hangzhou 311300, China; 4Animal Husbandry Technology Promotion Station of Zhejiang Province, Hangzhou 310021, China; vineus425@gmail.com (J.Y.); abiaosi@sina.com (H.G.); 5Key Laboratory of Feed Biotechnology of Ministry of Agriculture, Feed Research Institute, Chinese Academy of Agricultural Sciences, Beijing100081, China; tuyan@caas.cn; 6Department of Animal and Poultry Sciences, Virginia Polytechnic Institute and State University, Blacksburg, VA 24061, USA; rrwhite@exchange.vt.edu

**Keywords:** growth performance, weaning, concentrate dependent

## Abstract

**Simple Summary:**

Weaning calves from milk is a major process for young animals, and is also influential for farmers. Currently, beef cattle calves are always weaned by using the conventional method, where calves and cows are usually kept in the same pen for 90-180 days, mainly breastfeeding. But this may be negatively affecting the efficiency of the beef cattle industry due to the short-term appetite reduction, feed loss, and delayed growth in calves after weaning. Weaning strategies for beef cattle raised on milk have been minimally studied. In this study, we found that concentrate-dependent weaning can help to enhance the adaptation to the solid feed with advantages for average daily gain, dry matter intake, and feed efficiency. Early weaning did not significantly affect body height, body length, circumference of cannon bone, or circumference of the chest. Under the circumstances of this experiment weaning as soon as the calves can consume 750 g/d of concentrate was the best, which could provide useful information for future studies relating to performance of cattle.

**Abstract:**

This study was conducted to investigate the effects of weaning based upon different concentrate intake on growth performance, health, and antioxidant status of Southern Chinese Cattle. Thirty female calves were used in the trial. Weaning strategy was the primary variable of interest and treatments included weaning when a calf consumed 1000 (W1000), 750 (W750), or 500 (W500) g of starter for three consecutive days. All calves received colostrum within 4 h of birth and colostrum intake was monitored to ensure all calves consumed 1.5 L within 12 h of birth. Calves were then fed fresh milk up to 2 L/d for 7 days. Between d 7 and 13, calves were fed a liquid feed composed of 50% fresh milk, and 50% milk replacer (MR) at maximum rate of 3 L/d. For the remainder of the study, calves were provided 3 L/d MR. Concentrate (starter), and grass hay were available ad libitum starting on d 1. Milk, MR, starter, and hay intakes were recorded daily. Calf body weights and measurements were recorded at birth and every 4 wk until weaning. Average weaning ages were 48 ± 4.5, 58 ± 4.6, and 65 ± 4.8 for W500, W750, and W1000, respectively. Between wk 8 and wk 21, calves in the W500 and W750 treatments had higher (*p* < 0.05) starter intakes than calves in the W1000 treatment. Body height, body length, circumference of cannon bone, circumference of the chest did not differ with weaning strategy (*p* > 0.05). There was no difference in body weight (BW) and average daily gain (ADG) on wk 5 when any of the calves were weaned (*p* > 0.05). In wk 9, BW of calves in W1000 tended to be higher than that of W750 and W500 (*p* = 0.10). However, calves in W1000 lost their BW advantage in wk 13. Calves’ ADG was not different during the whole experiment period among treatments. Calves in W750 had higher plasma BHBA and the total antioxidant capacity which has been associated with a positive impact on health. Data in this experiment suggest that weaning when calves consume 750 g of starter can enhance ADG, DMI, feed efficiency, and selected blood parameters compared with weaning when calves consume 500 g of starter.

## 1. Introduction

The early weaning (EW) of beef calves is hypothesized to decrease feed and labor expenditures [1], shorten the postpartum resumption of reproduction [2], decrease voluntary forage intake, and increase dam pregnancy rates [3]. Blanco et al. [4] showed that, except for an improvement in dressing percentage, EW modified calf performance without affecting carcass characteristics. Moreover, public concern about animal welfare remains high in Europe, and is increasing in North America [5]. Now China also attaches great importance to animal welfare Vasseur et al. [5] indicated that weaning calves from milk is a major source of stress for young animals, and is also challenging for producers. Dairy calves are usually fed milk or milk replacer at a rate of 8% to 11% of body weight (BW) per day, and weaned at 45 to 60 d in a modern system or 90 to 180 d in small-holder system [6]. Although widely applied in the dairy industry, weaning strategies for beef cattle raised on milk have been minimally studied, largely because most production systems worldwide raise calves with cows. Yellow cattle calves are always weaned by using the conventional method at the age of 90–180 d [7]. However, this may be negatively affecting the efficiency of the beef cattle industry, mainly reflected in the high cost of liquid feed, reduced appetite in the short term, reduced feed intake, and delayed growth.

In Chinese systems, severe limitations on available forage mean that reducing cattle on pasture is of primary interest and, as a result, beef calves are often pulled from the dam much earlier than in other production systems. As such, improving our understanding of when and how to wean beef calves fed milk or milk replacer diets is of key importance. In particular, weaning based on age may not be the most consistent way to identify a calf’s ability to consume and digest solid feed. Zhang et al. [8] indicated that Holstein calves could be weaned when concentrate (starter) consumption reached 700 g/d. There is considerable interest within the Chinese beef industry to identify similar weaning time recommendations for beef calves raised on milk or milk replacer diets. 

The objective of this study was to investigate the effects of the concentrate consumption level required for weaning on performance, health, and antioxidant status of small breed Chinese beef cattle. In addition, as far as we know, there are few studies on weaning based on the amount of concentrate intake. We hypothesized that, much like the benefits seen with dairy calves, delaying weaning until calves were consuming at least 700 g of starter would result in improved growth, efficiency, and health. 

## 2. Materials and Methods

### 2.1. Animals, Diets, and Experimental Design

The use of animals was approved by the Animal Care Committee, Zhejiang A & F University, Hangzhou-Lin’an, China. Thirty healthy female beef calves (Southern Chinese Cattle) were stratified into 10 blocks of three cattle each according to date of birth and body weight. One calf per block was allocated to each treatment. Calves were weaned according to treatment when they consumed 1000, 750, or 500 g/d starter for three consecutive days (W1000, W750, and W500, respectively). As soon as a calf consumed the target quantity of starter for three consecutive days, the reduction of milk began with the threshold value being 0.5–1 L/d of milk based on the concentrate consumed. 

All calves received colostrum within 4 h of birth and colostrum intake was monitored to ensure all calves consumed 1.5 L within 12 h of birth. Calves were then trained to drink milk from buckets and fed up to 2 L/d of fresh milk until d 7. Milk feeding bottles were washed and disinfected with 5% hypochloride solution after each feeding according to the procedure of Khan et al. [9]. After d 7, calves were placed into individual pens and fed a liquid feed composed of 50% fresh milk and 50% milk replacer (MR) at maximum rate of 3 L/d. The MR contained 93.2% dry matter (DM), 23.9% crude protein (CP; DM basis), and 13.1% ether extract (DM basis; Patent No: ZL02 128844.5). At each feeding (2× daily), the MR powder was dissolved in water (approximately 38 °C) at a weight to volume ratio of 1:7. The liquid feed mixture was offered to calves from d 7 to 13, and was increased gradually (by 0.5 L/d) until the maximum allowance of 3 L/d was reached. After d 13, calves were fed 3 L/d liquid MR for the remainder of the study until the weaning. This procedure was chosen in this study because it provided a consistent artificial rearing strategy that could be replicated across calves with minimal variation. 

A pelleted (6 mm diameter and 10 mm long) starter was offered twice daily at 09:00 and 16:00 from 10 d to 3 months of age. The ingredients of the starter were 58.5% corn, 27.0% soybean meal, 10.0% wheat bran, 1.0% salt, 1.1% dicalcium phosphate, 1.4% limestone, and 1.0% vitamin/mineral premix. The starter contained 17.9% CP, 10.9 MJ metabolic energy (ME)/kg DM, 20.7% neutral detergent fiber (NDF), 4.5% acid detergent fiber (ADF), 0.8% Ca, and 0.6% P. When calves were 3 months old, they were transitioned to a finishing diet, which they received for the duration of the study. The ingredients of finishing diet were 62.4% corn, 22.0% soybean meal, 11.0% wheat bran, 1.0% salt, 1.3% dicalcium phosphate, 1.3% limestone, and 1.0% vitamin/mineral premix, respectively. The chemical composition for finishing diet was 16.3% CP, 10.9 MJ ME/kg DM, 13.4% NDF, 4.2% ADF, 0.8% Ca, and 0.6% P. In both the calf starter and the finishing ration, the premix inclusion rate was designed so the diet contained 15,000 IU Vitamin A, 5000 IU Vitamin D, 50 mg Vitamin E, 90 mg Fe, 1.5 mg Cu, 60 mg Mn, 100 mg Zn, 0.3 mg Se, 1.0 mg I, and 0.5 mg Co, all expressed per kg diet. The quantity of concentrate provided was adjusted daily to target 5% to 10% refusals. All calves were offered ad libitum access to grass hay and clean water from a plastic bucket throughout the study. Given assumed grass hay intakes, the concentrate mixtures were formulated to meet the nutrient requirements of beef cattle according to the Chinese feeding standard [10] and data of Diao and Tu [6]. 

### 2.2. Sampling, Measurement, and Analysis

The amounts of milk, MR, starter, and hay intakes were recorded daily. The diet was sampled weekly and samples were dried in a forced-air oven at 60 °C for 48 h and composited for analysis of DM, CP, Ca [11], and P [12]. Samples were also analyzed for ADF and NDF according to the procedure described by Van Soest et al. [13]. Total dry matter intake (DMI) was defined as the sum of consumed MR, concentrate and hay on a dry basis.

Body weights and measurements were collected at birth and every 4 wk thereafter until the end of the study. Body measurements included body weights, body height, body length, circumference of cannon bone, and circumference of the chest. Calf average daily gain (ADG) was calculated based on the difference in body weight (BW) over time. Feed efficiency (FE) was calculated for wk 5 to 9 and wk 9 to 21, as daily DMI (MR + concentrate + hay) divided by ADG. Blood samples (10 mL) were taken from the coccygeal vein at the beginning of wk 7, 9, and 21. Samples were collected prior to feeding and were immediately transferred into heparinized tubes. The samples were centrifuged at 3000× g for 10 min to separate plasma and plasma was analyzed for glucose (GLU) [14]; blood urea nitrogen (BUN) [15]; total protein (TP) [16]; β-hydroxybutyric acid (BHBA) [16]; glutathione peroxidase (GSH-P_X_) [17]; superoxide dismutase (SOD) [17]; malondialdehyde (MDA) [17]; and total antioxidant capacity (T-AOC) [18] by Nanjing Jiancheng Bioengineering Institute (Nanjing, China). 

Health status was monitored daily by research staff and diseases were diagnosed by an experienced veterinarian. Fecal score was also recorded weekly from weaning until the end of the study. A 4-point scale was used according to Alugongo’s method with scores of 1 = firm, 2 = slightly loose, 3 = loose, and 4 = watery [19].

### 2.3. Statistical Analysis

All data except for weaning age (WA), feed efficiency (FE), and incidence of diseases were analyzed as a randomized complete block design using PROC MIXED of SAS (version 9.2; SAS Institute Inc., Cary, NC, USA). The model included treatment, time, and interaction of treatment × time as fixed effects, and calf within treatment as a random effect. Time was used as a repeated measure and the autoregressive (1) error structure was selected based on AIC and BIC. To better account for animal-to-animal differences the birth BW values were included as a covariate. Block was deleted because it was not significant for any response variable. Results are reported as least squares means. Mean separation was conducted using the PDIFF option. Data for WA and FE were analyzed using the Proc GLM of SAS because they were not repeated within animal. The statistical model used only treatment as a main effect and calf BW as a covariate. Data on FE were separately analyzed for wk 5 to 9 (weaning period) and wk 9 to 21 (post weaning period). Incidence of diseases were analyzed for treatment effects using the FREQ procedure of SAS with a chi-square test and Fisher’s exact test. For all analyses, differences were considered significant when *p* was <0.05 and as trends when 0.05 ≤ *p* ≤ 0.10.

## 3. Results

### 3.1. Weaning Age, Feed Intake, and Growth Performance

Initial BW of the calves did not differ among treatments (Table 1; *p* = 0.84); however, weaning strategy did affect mean WA. As the required cut off concentrate provision level increased from 500 to 1000 g, WA increased from 48 ± 4.5 to 65 ± 4.8 d (Table 1). 

Intake of MR, starter, and hay is presented in Figure 1. Calves attained milk intakes of 3 L/d in about 2 wk, and were fed 3 L/d of liquid MR for the remainder of the study until the weaning. No difference of starter intake was found before wk 8 (*p* > 0.05), but starter intakes in W1000 were persistently lower than in W750 group from wk 9 through the end of the study (Figure 1). Hay intake was low from wk 1 to 5 with an average DMI of 29.9 ± 0.23 g/d. Intake of hay progressively increased from wk 4 until the end of the study, but weaning method did not affect hay intake (*p* > 0.05). From wk 5 to 9, milk intake of the W1000 group was significantly higher than that of the other two groups (*p* < 0.01; Table 1). Although concentrate intake was not affected by treatment before week 9 (*p* = 0.19), concentrate intake of W1000 was significantly lower than that of theW750 group between wk 9 and 21 (*p* < 0.05).

The BW change over time for each treatment is shown in Figure 2. In wk 5, prior to weaning, no significant differences in BW were found associated with treatment (*p* > 0.05). By wk 9, BW of calves on the W1000 treatment were numerically (*p* = 0.12) higher than that of calves on the W750 and W500 treatments (42.9 ± 1.64 kg vs. 39.0 ± 1.62 and 40.0 ± 1.74 kg). However, this difference in BW was no longer significant by wk 13 (Figure 2; *p* > 0.05). In wk 21, BW of W1000 was less than BW of either W750 or W500 (*p* < 0.05) (95.6 ± 1.75 and 93.1 ± 1.62 kg vs. 87.6 ± 1.64 kg). The BW of W750 and W500 were not different (*p* > 0.05).

As shown in Table 1, ADG from wk 1 to 21 was 0.495, 0.552, and 0.533 for groups W1000, W750, and W500, respectively, with no treatment effects observed (*p* > 0.05). When ADG was analyzed for each 4-week period, no treatment effects were found except for wk 9 to 13 (Table 1). In this period (wk 9 to 13), the ADG of the W1000 group was lower (0.309 kg/d) than that of the W750 (0.577 kg/d) and W500 (0.581 kg/d) groups (*p* < 0.05). 

Calf FE responses to weaning strategy were not significantly different prior to wk 9. After wk 9, the W1000 group had a lower (*p* < 0.05) FE than the other two weaning strategies. Despite differences in FE, BW, and ADG, body measurements did not differ among groups in wk 21 compared with that of W750 (Table 2).

### 3.2. Blood Metabolites

As shown in Table 3, there was no effect of treatment on BUN, GLU, or TP concentrations (*p* > 0.05). In wk 7, concentration of BHBA did not differ among groups, but by wk 9, the calves on the W750 treatment had higher plasma BHBA than that of W1000 (*p* < 0.05). This trend continued in wk 21 with W750 calves tending to have higher plasma BHBA than B1000 calves (*p* > 0.05). Plasma BHBA did not differ between calves on the W750 and W500 treatments (*p* > 0.05).

Effects of concentrate-dependent weaning methods on blood antioxidant defense parameters are shown in Table 4. In wk 7, weaning treatment did not affect plasma MDA, GSH-Px, or T-AOC (*p* > 0.05). However, blood SOD activity for calves on the W500 treatment was lower than for calves on the W750 or W1000 treatments in wk 7 (*p* < 0.05). This difference in SOD activity associated with weaning strategy was no longer significant in wk 21 (*p* > 0.05). There was no significant difference in plasma MDA concentration among treatments. The T-AOC activity increased with calf age (*p* < 0.05). Additionally, in wk 9, calves on the W1000 treatment had lower T-AOC activity (*p* < 0.05) than calves receiving the other weaning treatments. This T-AOC difference was not apparent in wk 7 or wk 21 (*p* > 0.05). Blood GSH-Px activity increased (*p* < 0.05) as calves aged, with the highest activity in wk 7. Calf plasma GSH-Px activity (Table 4) did not differ with weaning method.

### 3.3. Health Status

Calves were generally healthy throughout the study. Only five calves were identified with diarrhea (two on W500, one on W750, and two on W1000), and these calves recovered within 3 to 4 days. Weaning method did not affect the incidence of diarrhea (*p* > 0.05) or fecal scores (*p* > 0.05). No other diseases occurred during the experiment period.

## 4. Discussion

### 4.1. Weaning Age

Most research on calf feeding and weaning management has been conducted in dairy calves, thus information is limited for beef cattle. Although a plethora of dairy calf work has defined strategies for feeding and weaning to optimize health and productivity, this is not directly translatable to beef cattle because of the differences in intake and body weight. For example, calves used in this experiment were Southern Chinese Cattle which have mature BW of 480 to 600 kg [20], which is much lower than the typical mature BW of Holstein cattle (≈680 kg). These differences in mature BW also convey differences in birth weight and feed intake, which complicate attempts to use Holstein-derived feeding recommendations for EW of beef calves. Application of EW strategies in the Chinese beef industry is further complicated because Southern Chinese Cattle have Bos indicus genetics meaning that they mature and respond to feeding strategies differently than cattle with only Bos tarus genetics. Additionally, improving our understanding of how to feed Southern Chinese Cattle early in life may help improve the efficiency of management of these cattle, which is currently a nationally recognized problem [20].

The need to evaluate Southern Chinese Cattle weaning strategies is exemplified by comparing literature based on Holstein calves with the values obtained in the present study. Calves in the W500 group were weaned at 48 d; however, Quigley [21] suggested the Holstein calves consumed 366 to 500 g of concentrate by 35 d. Similarly, Ghorbani et al. [22] indicated that by 58 d, Holstein calves were consuming 900 g/d of starter. In this study, calves did not consume 1000 g/d starter until 65 d. Differences in birth weight, mature weight, and DMI among the cattle breeds likely contributed to these differences in ages. In a different study [23], WA for conventionally reared calves was 84 d, but WA for calves weaned based on concentrate intake was 76 d. This study is not directly comparable to our experiment because calves were weaned when they consumed 2000 g of concentrate per day.

Irrespective of production system, WA is commonly used as a criteria for weaning. A lot of (32.9%) producers weaned calves at 56 d of age; a small group (2.3%) of producers weaned calves as early as 21 d; and some (14.2%) producers delayed weaning for ≥12 wk [21]. Another report by USDA National Animal Health Monitoring and Surveillance indicated that the average WA was 8.2 wk of age, and for large operations the average WA was 9.1 wk [24]. The normal WA for beef calves reared in typical U.S. cow-calf production systems was even much later, with age at 200 to 300 d [25,26,27]. Although early weaning work in beef cattle has tested weaning at much younger ages (CITE EARLY WEANING STUDIES), there is interest in China in weaning calves as early as possible due to limited forage resources to support lactating cows. The present study suggested that applying feed consumption targets applied in the dairy industry could facilitate EW of beef calves at ages much younger than previously investigated. Calves in W500 were weaned between 43 and 54 d and calves in W1000 were weaned between 54 and 68 d. As reported by Vasseur et al. [5], the median age at weaning was 49 d, the median BW was 82 kg, and the median concentrate intake was 2 kg. Additionally, WA could be much earlier than usual in the research of Li and Diao [28], and calves weaned at 10 and 20 d had higher ADG compared with that of weaned at 70 d of age. Similarly, Hulbert et al. [1] found no difference on ADG of calves weaned when they consumed 900 g/d starter (23.7 d) or conventional method (44.7 d). In contrast, in organic farms, the WA, BW, and concentrate intake was 196 d, 220 kg, and 4.5 kg, respectively [5]. Moreover, WA could be reduced by replacing the whole milk by 25% to 50% of soymilk [22].

### 4.2. Growth Performance

The BW and ADG of calves in this study were slightly lower than values in studies using larger breeds of cattle [23,25,29]. The ADG over the feeding period was 0.483 kg/d, which was 90.4% lower than reported by Grings et al. [29] for crossed breed cattle. The difference in ADG may be due to differences in birth weight. Calves in the current study had birth weights that were 40.5% of the birth weights reported by Grings et al. [29]. Although the average BW and ADG were lower than those described in previous studies, these differences are likely attributable to breed and mature weight differences.

Changes in ADG during the weaning period identified in this study were inconsistent with previous work. For example, Hulbert et al. [1] found that ADG after weaning tended to be higher in calves that were EW. In our experiment, ADG did not differ among treatments. It is possible that this lack of difference was due to a limited weaning effect on calf performance. Weaning is often associated with reduced weight gain in some calves, which may be due to the insufficient intake of concentrate at the time milk feeding is stopped [23]. In this study, we observed a slight weaning effect. After weaning (wk 9) BW of calves in W1000 tended to be higher than that of calves on the W750 and W500 treatments. However, these differences were no longer significant by wk 13 suggesting that the calves on the W500 and W750 treatments compensated for the initially lower ADG. An alternative reason for the lack of ADG differences by wk 13 is the reduced intake of concentrate in the W1000 group. Calves adapting to starter on the W1000 treatment increased starter intake at a lesser rate than the W500 and W750 calves (Figure 1). The depressed rate of increase in starter intake of the W1000 calves suggests that these calves did not adapt as easily to starter as the calves weaned at W500 and W750. It is possible that calves have been drinking longer and their rumens were not as developed as those calves that were transitioned off milk earlier in life. We found that after weaning in the eighth week, the calves did not get enough nutrients to achieve higher ADG. In fact, concentrate may replace milk’s effect of promoting the growth of BW, but it was negatively affected by the weaning in W1000. The lower ADG was also consistent with the lower FE in W1000 compared with that of other two groups. As reported by Roth et al. [23], the concentrate consumption by concentrate dependent weaning group was 11.7% higher than that of the conventional weaning group, and consequently the weight gain was 11.1% higher for the concentrated-dependent weaning group.

### 4.3. Blood Metabolites

Quigley et al. [21] found that concentration of BUN was greater for calves gradually weaned than that of calves that were abruptly weaned. In this study, no differences in BUN and TP concentrations were found to be associated with weaning strategy. No difference in GLU concentration was found among treatments or as calves aged. In contrast to this finding, Zhang et al. [8] indicated that the concentration of plasma GLU decreased significantly with advancing age of calves. The difference between Zhang et al. [8] and this study may be due to the different monitoring period (3 to 13 wk in Zhang et al. [8] vs. 5 to 21 wk in the present study). The longer monitoring period in this study may have diluted any age-related effects in GLU.

Plasma BHBA concentration increased approximately 1.27 to 2.50 fold during the weaning period. The increase in plasma BHBA concentration as calves aged was similar to results reported by other researches [21,30]. In previous work, the decrease in blood concentrations of GLU and increase in blood BHBA has been associated with the shift in energy sources during the transition from liquid to solid diets [9].

No treatment differences in plasma BHBA were found before weaning (wk 7); however, after weaning (wk 9 to 21) BHBA concentration was significantly higher in the W750 treatment and numerically higher in the W1000 treatment compared with that of the W500 treatment. These changes were expected because plasma BHBA concentration is tightly correlated with concentrate intake [8]. Plasma BHBA is also a measure of rumen epithelial metabolic activity and indicates conversion of rumen butyrate to β-hydroxybutyrate as it passes the rumen wall [9]. The lower plasma BHBA concentration for group W1000 was probably due to a lower concentrate intake during the post-weaning period.

Antioxidant status was evaluated by measuring blood SOD, MDA, GSH-Px, and T-AOC levels in the present study. The SOD is known as an antioxidant enzyme that promotes the conversion of an anion superoxide to H_2_O_2_ [31], and GSH-Px prevents free radical damage to phospholipid membranes, enzymes, and other important molecules [32]. High level of MDA in dermatophytic calf blood usually indicates an advanced peroxidative process in cell membranes [33]. Ranade et al. [34] indicated that the development of antioxidant capacity took place over time and there may well be some reduced antioxidant capacity in the first 3 months of the life as the calf develops. In agreement with that reported by Ranade et al. [34], we observed a decline in plasma GSH-Px with advancing of calf age. Ranade et al. [34] indicated that concentrations of H_2_O_2_ in exhaled breath condensate at weaning were significantly higher than those measured before weaning. The main function of the antioxidant defense system is to maintain the concentration of free radicals and oxygen active forms at a stable level. Therefore, the lower SOD activity for calves of W500 in wk 7 of the present study may be a result of oxidant status. This is also supported by the lower plasma MDA for the W500 group in wk 7. The higher concentration in plasma MDA concentration in W1000 may be a result of increased oxidative stress and lower plasma T-AOC in this group of calves. In this case, it may be hypothesized that calves of W1000 in the current study were under oxidative stress (as indicated by increased blood MDA), and, consequently, a state of negative control on T-AOC. This elevated oxidative stress may have partially contributed to the depressed starter intake of this group and be partially responsible for the lack of differences in ADG among calves.

## 5. Conclusions

In summary, concentrate-dependent weaning can help to enhance the adaptation to the solid feed with advantages on ADG, DMI, and FE. Early weaning did not significantly affect body height, body length, circumference of cannon bone, and circumference of the chest. Moreover, late weaning may cause a reduction in plasma BHBA and the T-AOC, with a lower ability to reduce oxidative compounds which has a negative impact on health. Under the circumstances of this experiment weaning as soon as the calves can consume 750 g/d of concentrate was the best. In the weaning management of beef calves, we should not only pay attention to the age, but also consider the intake of the calves.

## Figures and Tables

**Figure 1 animals-10-00018-f001:**
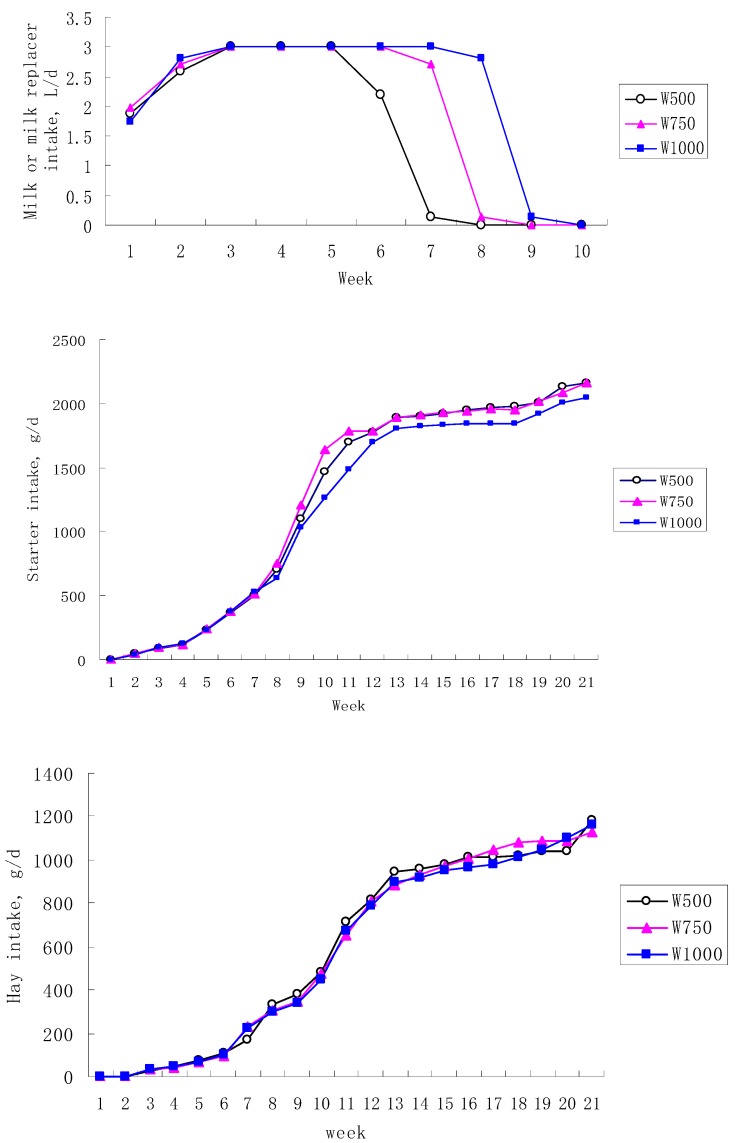
Mean milk, starter, and hay intake for 500, 750, and 1000 g/d concentrate-dependent weaning calves at each week of the experiment.

**Figure 2 animals-10-00018-f002:**
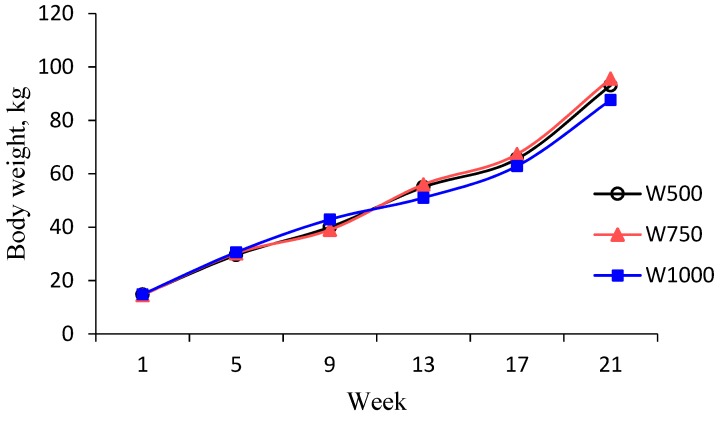
Least squares means for body weight of calves weaned when consumption of concentrate at 500, 750, and 1000 g/d.

**Table 1 animals-10-00018-t001:** Effects of concentrate-dependent weaning methods on weaning ages and performance of Southern Chinese Cattle calves.

Item	Concentrate-Dependent Weaning ^1^			SEM	*p* ^2^		
	W1000	W750	W500		T	D	T*D
Average weaning age, d	65 ^a^	58 ^b^	48 ^c^	1.8	<0.01	-	-
Initial BW, kg	14.9	14.6	14.7	0.377	0.84	-	-
Average daily gain, kg/d							
wk 1 to wk 21	0.495	0.552	0.533	0.047	0.13	<0.01	0.07
wk 1 to wk 5	0.524	0.522	0.495	0.062	0.72	-	-
wk 5 to wk 9	0.401	0.301	0.300	0.053	0.23	-	-
wk 9 to wk 13	0.309 ^b^	0.577 ^a^	0.581 ^a^	0.042	0.02	-	-
wk 13 to wk 17	0.678	0.702	0.658	0.060	0.73	-	-
wk 17 to wk 21	0.630	0.738	0.719	0.045	0.26	-	-
wk 5 to 9 of age							
Milk intake, g/d	341.1 ^a^	252.6 ^b^	152.6 ^c^	20.3	<0.01	<0.01	0.41
Concentrate intake, g/d	550.0	618.4	580.0	28.1	0.19	<0.01	0.34
Hay intake, g/d	205.5	219.2	214.2	30.8	0.34	<0.01	0.69
Feed: gain, kg of DM/kg of BW	2.67	2.66	3.03	0.20	0.22	-	-
wk 9 to 21 of age							
Concentrate intake, g/d	1730.0 ^b^	1866.5 ^a^	1841.6 ^ab^	45.0	0.05	0.04	0.42
Hay intake, g/d	868.5	885.2	889.1	39.0	0.43	<0.01	0.46
Feed: gain, kg of DM/kg of BW	4.83 ^a^	4.08 ^b^	4.18 ^ab^	0.31	0.03	-	-

^a, b^ Means in the same row with different superscripts differ significantly (*p* < 0.05). ^1^ W1000, W750, and W500 indicated that calf was weaned when it consumed 1000, 750, or 500 g of starter, respectively. ^2^ T = effect of treatment; D = effect of time; and T*D = interaction effect of treatment and time.

**Table 2 animals-10-00018-t002:** Effects of concentrate-dependent weaning methods on body measurements of Southern Chinese Cattle calves.

Item	Concentrate-Dependent Weaning ^1^			SEM	*p* ^2^		
	W1000	W750	W500		T	D	T*D
Body height, cm	68.8	70.0	69.2	0.50	0.17	<0.01	0.34
Body length, cm	64.3	66.7	65.0	0.70	0.08	<0.01	0.62
Circumference of cannon bone, cm	9.89	10.1	9.83	0.200	0.53	<0.01	0.37
Circumference of the chest, cm	86.8	90.2	89.0	0.83	0.34	<0.01	0.47

^a, b^ Means in the same row with different superscripts differ significantly (*p* < 0.05). ^1^ W1000, W750, and W500 indicated that calf was weaned when it consumed 1000, 750, or 500 g of starter, respectively. ^2^ T = effect of treatment; D = effect of time; and T*D = interaction effect of treatment and time.

**Table 3 animals-10-00018-t003:** Effects of concentrate-dependent weaning methods on blood metabolites of Southern Chinese Cattle calves.

Item ^2^	Concentrate-Dependent Weaning ^1^			SEM	*p*
	W1000	W750	W500		
BUN, mmol/L					
Wk 7	6.76	7.02	7.17	0.66	0.67
Wk 9	6.32	7.46	7.60	0.76	0.33
Wk 21	6.80	7.84	7.08	0.70	0.22
TP, g/L					
Wk 7	67.8	73.7	70.1	2.70	0.22
Wk 9	66.3	69.6	67.3	2.33	0.29
Wk 21	69.2	74.1	70.1	3.43	0.46
GLU, mmol/L					
Wk 7	4.18	4.13	3.66	0.32	0.39
Wk 9	4.58	4.23	4.44	0.34	0.43
Wk 21	4.30	3.89	3.80	0.40	0.54
BHBA, µmol/L					
Wk 7	273.9	232.5	213.9	25.8	0.48
Wk 9	460.9 ^b^	596.8 ^a^	533.3 ^ab^	33.7	0.03
Wk 21	584.2	746.5	658.7	58.9	0.15

^a, b^ Means in the same row with different superscripts differ significantly (*p* < 0.05). ^1^ W1000, W750, and W500 indicated that calf was weaned when it consumed 1000, 750, or 500 g of starter, respectively. ^2^ BUN: blood urea nitrogen, TP: total protein, GLU: glucose, GSH-Px: glutathione peroxidase, NEFA: non-esterified fatty acid, BHBA: β-hydroxybutyric acid, SOD: superoxide dismutase, MDA: malondialdehyde, T-AOC: total antioxidant capacity.

**Table 4 animals-10-00018-t004:** Effects of concentrate-dependent weaning methods on blood antioxidant defense parameters of Southern Chinese Cattle calves.

Item ^2^	Concentrate-Dependent Weaning ^1^			SEM	*p*
	W1000	W750	W500		
SOD, U/mL					
Wk 7	144.5 ^a^	139.3 ^ab^	117.6 ^b^	4.19	0.02
Wk 9	136.5	121.9	118.0	6.30	0.29
Wk 21	124.8	139.3	130.0	10.7	0.47
MDA, nmol/mL					
Wk 7	2.33	2.36	2.76	0.23	0.26
Wk 9	2.99	2.43	2.28	0.27	0.18
Wk 21	2.67	2.51	2.55	0.25	0.46
T-AOC, U/mL					
Wk 7	2.96	3.21	2.34	0.31	0.39
Wk 9	2.12 ^b^	3.27 ^a^	2.77 ^ab^	0.30	0.04
Wk 21	3.76	4.07	3.86	0.18	0.61
GSH-Px, U/mL					
Wk 7	100.0	104.7	118.5	6.10	0.54
Wk 9	47.9	56.1	51.2	2.91	0.28
Wk 21	69.9	77.7	75.1	4.10	0.82

^a, b^ Means in the same row with different superscripts differ significantly (*p* < 0.05). ^1^ W1000, W750, and W500 indicated that calf was weaned when it consumed 1000, 750, or 500 g of starter, respectively. ^2^ GSH-Px: glutathione peroxidase, SOD: superoxide dismutase, MDA: malondialdehyde, T-AOC: total antioxidant capacity.

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
