# Peer review of "Weaning Performance of Beef Cattle Calves Based on Concentrate Intake"

_animals, 2019, doi:10.3390/ani10010018_

Round 1
Reviewer 1 Report
Review of Manuscript Animals-641437
The paper aimed to evaluate the effect of weaning strategies based on different concentrate intakes on growth performance, health and blood parameters of Yellow Chinese Beef calves. In general the paper fits with the aim and scope of the journal and the information presented in this paper is quite interesting and relevant especially for beef cattle farmers and researchers in China. I just would recommend authors to consider the following comments to improve the paper
General comments
The problematic was properly introduced and authors highlighted the importance of the study. Authors properly mentioned the objectives and stated clear hypothesis.
The M&M is very well structured and clearly described with enough detail of all materials and methods. Authors must just clearly state whether they determined the feed intakes of animals individually or not. This is of great relevance.
The chapter results provides a precise and concise description of the most important results. If the journals allows it, I would emphasise and recommend authors in presenting the exact p-values they obtained in the text and not using just p> or p< 0.05. By this, each reader can interpretate the values by its own.
The discussion of results is clear and scientifically based and supported with other studies from the literature. I would recommend authors to mention some directions and recommendations for future research attempting to improve weaning management of yellow beef cattle in China.
Specific comments
Title
I would avoid in the title using the word weaning/weaned (twice). Delete the second one weaned.
Simple summary/Abstract
L16: using the conventional method based on…. (mention the method to make it clear)
L18: industry because… (mention the reason to make it clear)
L18: I doubt that the strategies have been minimally studied. Indeed there is quite good literature. Or do you mean under Chinese conditions and especially for the studied breed? If yes, please specify
L21: Body measurements is too general. Be specific
L33: Intakes were recorded individually or per group?
L34-37: I think statistical analyses methods are not required in the abstract. Make emphasis on showing results and discussing them.
L40: Body measurements? Be specific
Introduction
L56: Is this concern about animal welfare also of interest in China? If not limit your statements to China
L62: What is this conventional method?
L63: How can this affect negatively the industry? Shortly explain
L70: 700 g/d, not kg/d
L73: Give the exact and complete name of the breed here considered. Is this the Southern Yellow Chinese Cattle breed?
L74: check letter size
Materials and methods
L101: 10.9 MJ ME/kg DM
L105: 10.9 MJ ME/kg DM
L110: Briefly provide the kind of grass hay and chemical composition
Here is not clear whether concentrate intake was determined per group or individually. Please specify, because this is very important
L115: Individually recorded? Very important. And how? Briefly explain
L120: What kind of measurements? Specify here together with body weight which is also a body measurement
L122: Over what times? Only initial and final weight? Specify
Result
L151: Preferable write the p value you observed and wrote in the table (0.84). See other instances
L162: But according Table 1 concentrate intake was statistically similar between W1000 and W500, based on superscripts. Maybe I misunderstood? Please Check
L165: Give p value = 0.34. Or do you mean both periods (wk 5-9 and wk 9-21)? Specify
L166: P<0.01
L167: P=0.19
L168: Not really, based on superscripts, concentrate intake of W100 and W500 are similar. Check
L176: Be careful. You stated in L147-148 that trends are p values between 0.05 and 0.10. Here you have a p-value of 0.12
L187: correct the word “group’
L199-200: These abbreviations should be introduced in L127-128
L212: Introduce these abbreviations in L128
Discussion/Conclusions
L232: Discussion in singular
L234-246: Very good justification
L252: use the abbreviation DMI
L257. Do not start a sentence with a number
L257-259: Is that for China and the breed you used? Is not clear. Specify that that was for Jersey. Comparable to your conditions?
L259-261: Specify here again conditions and breed. Make clear when you use data from Holstein or other breed calves or beef calves. Consider this in other circumstances along the manuscript
L297-301: The sentence is too long. Dived it
L308-309: Was the because of gradual reduction of milk intake?
L341: The higher concentration in plasma…
L347: Check the word conclusion
L350: specify the body measurements
Author Response
Response to Reviewer 1 Comments
Review of Manuscript Animals-641437
The paper aimed to evaluate the effect of weaning strategies based on different concentrate intakes on growth performance, health and blood parameters of Yellow Chinese Beef calves. In general the paper fits with the aim and scope of the journal and the information presented in this paper is quite interesting and relevant especially for beef cattle farmers and researchers in China. I just would recommend authors to consider the following comments to improve the paper
We appreciate the comments and suggestions, the point-to-point responses are shown below.
General comments
The problematic was properly introduced and authors highlighted the importance of the study. Authors properly mentioned the objectives and stated clear hypothesis.
Author response:Thank you
The M&M is very well structured and clearly described with enough detail of all materials and methods. Authors must just clearly state whether they determined the feed intakes of animals individually or not. This is of great relevance.
Author response: These calves were raised individually, intakes were recorded individually
The chapter results provides a precise and concise description of the most important results. If the journals allows it, I would emphasise and recommend authors in presenting the exact p-values they obtained in the text and not using just p> or p< 0.05. By this, each reader can interpretate the values by its own.
Author response: Thanks for your comment. We marked the significance according to the format of the journal.
The discussion of results is clear and scientifically based and supported with other studies from the literature. I would recommend authors to mention some directions and recommendations for future research attempting to improve weaning management of yellow beef cattle in China.
Author response: we added it in the conclusion.
Specific comments
Title
I would avoid in the title using the word weaning/weaned (twice). Delete the second one weaned.
Author response: Revised as suggested.
Simple summary/Abstract
L16: using the conventional method based on…. (mention the method to make it clear)
Author response: This part has been revised.
L18: industry because… (mention the reason to make it clear)
Author response: This part has been revised.
L18: I doubt that the strategies have been minimally studied. Indeed there is quite good literature. Or do you mean under Chinese conditions and especially for the studied breed? If yes, please specify
Author response: In China, dairy cattle breeding management has become systematic and standardized, but there is little reference for beef cattle breed. At present, weaning methods of meat calves in China are uneven. Compared with milk calves, the survival rate is lower and the incidence is higher. Prior to this experiment, Guo, F, et al. have done research on weaning of beef calves(reference 7).
L21: Body measurements is too general. Be specific
Author response: This part has been revised.
L33: Intakes were recorded individually or per group?
Author response: Intakes were recorded individually.
L34-37: I think statistical analyses methods are not required in the abstract. Make emphasis on showing results and discussing them.
Author response: Thanks for your suggestion,statistical analyses methods were deleted.
L40: Body measurements? Be specific
Author response: This part has been revised.
Introduction
L56: Is this concern about animal welfare also of interest in China? If not limit your statements to China
Author response: Yes it is. We had added it to the manuscript (L59).
L62: What is this conventional method?
Author response: Traditional beef cattle breeding method generally raises calves and cows in the same pen for 90-180 days, mainly breastfeeding.
L63: How can this affect negatively the industry? Shortly explain
Author response: The explaination was added to this section.
L70: 700 g/d, not kg/d
Author response: The typo was corrected.
L73: Give the exact and complete name of the breed here considered. Is this the Southern Yellow Chinese Cattle breed?
Author response: Yes it is.
L74: check letter size
Author response: This part has been revised.
Materials and methods
L101: 10.9 MJ ME/kg DM
Author response: This part has been revised.
L105: 10.9 MJ ME/kg DM
Author response: This part has been revised.
L110: Briefly provide the kind of grass hay and chemical composition
Author response: We used chinese Leymus chinensis, the nutritional composition of Leymus chinensis is 8.2% CP, 2.71% EE, 4.35% ash, 32.41% CF, 42.83% NFE, 0.519% Ca, and 0.088% P.
Here is not clear whether concentrate intake was determined per group or individually. Please specify, because this is very important
Author response:We recorded individual concentrate intake daily.
L115: Individually recorded? Very important. And how? Briefly explain
Author response:Record the amount of individual feed and the remaining amount every day by experimenter. We prepared electronic scales.
L120: What kind of measurements? Specify here together with body weight which is also a body measurement
Author response: This part has been revised.
L122: Over what times? Only initial and final weight? Specify
Author response: This experiment lasts 21 weeks. We gained six weights (1wk initial weight, 5wk, 9wk, 13wk, 17wk, 21wk).
Result
L151: Preferable write the p value you observed and wrote in the table (0.84). See other instances
Author response: Revised as suggested.
L162: But according Table 1 concentrate intake was statistically similar between W1000 and W500, based on superscripts. Maybe I misunderstood? Please Check
Author response: According Table 1 concentrate intake was statistically similar between W1000 and W500, but significant difference between W1000 and W750 groups. Thanks for your question, this part was revised.(L171).
L165: Give p value = 0.34. Or do you mean both periods (wk 5-9 and wk 9-21)? Specify
Author response: According Table 1, weaning method did not affect hay intake, both two periods (wk 5-9 and wk 9-21). P(wk 5-9)=0.34, P(wk9-21)=0.43.
L166: P<0.01
Author response: Revised as suggested.
L167: P=0.19
Author response: Revised as suggested.
L168: Not really, based on superscripts, concentrate intake of W100 and W500 are similar. Check
Author response: Concentrate intake was statistically similar between W1000 and W500, but significant difference between W1000 and W750 groups. Thanks for your question, this part was revised.(L177).
L176: Be careful. You stated in L147-148 that trends are p values between 0.05 and 0.10. Here you have a p-value of 0.12
Author response: This part was revised.
L187: correct the word “group’
Author response: The typo was corrected. (L195).
L199-200: These abbreviations should be introduced in L127-128
Author response: Revised as suggested. (L133-L136).
L212: Introduce these abbreviations in L128
Author response: Revised as suggested. (L133-L136).
Discussion/Conclusions
L232: Discussion in singular
Author response: Revised as suggested.
L234-246: Very good justification
Author response: Thank you.
L252: use the abbreviation DMI
Author response: Revised as suggested. (L261)
L257. Do not start a sentence with a number
Author response: Thanks for your suggestion, the sentence was revised. (L266)
L257-259: Is that for China and the breed you used? Is not clear. Specify that that was for Jersey. Comparable to your conditions?
Author response: The breeds were different from our study. This is just a comparison of an average value in the United States.
L259-261: Specify here again conditions and breed. Make clear when you use data from Holstein or other breed calves or beef calves. Consider this in other circumstances along the manuscript
Author response: In the literature, crossbred cattle were used. Thanks for your suggestion. It is really important to pay more attention to breed.
L297-301: The sentence is too long. Dived it
Author response: Revised as suggested.
L308-309: Was the because of gradual reduction of milk intake?
Author response: In our opinion, if milk intake decreases, calves may need more pellets and roughage to sustain their own development. On the one hand, the intake is increased, and on the other hand, solid feed can promote the improvement of rumen function. Therefore concentrated consumption is higher.
L341: The higher concentration in plasma…
Author response: Revised as suggested.
L347: Check the word conclusion
Author response: The typo was corrected.
L350: specify the body measurements
Author response: Revised as suggested.

Reviewer 2 Report
To authors
I have reviewed your manuscript. The manuscript is written well. Detail comments are as follows:
L70 700kg→700g
L125 Describe the reason of the blood collection period (7, 9 and 21 wks) are not same to BW measurement (1, 5, 9, 13, 17 and 21wks)
L152 concentrate concentration→concentrate provision level or concentrate allowance
Table 1 Is the value of "Average daily gain" geometric mean value?
Check the Item (delete ee and wk 5 to 9 of age)
Author Response
Response to Reviewer 2 Comments
I have reviewed your manuscript. The manuscript is written well. Detail comments are as follows:
L70 700kg→700g
Author response: The typo was corrected.
L125 Describe the reason of the blood collection period (7, 9 and 21 wks) are not same to BW measurement (1, 5, 9, 13, 17 and 21wks)
Author response: We collected blood samples at 7 weeks, 9 weeks, and 21 weeks to detect the effects of weaning methods on blood indicators. By 7 weeks, each experimental group was in the weaning stage, by 9 weeks, each experimental group was in the post-weaning period. Body weight is measured every four weeks to observe the growth performance of calves at various stages, and feed conversion rate.
L152 concentrate concentration→concentrate provision level or concentrate allowance
Author response: Revised as suggested.
Table 1 Is the value of "Average daily gain" geometric mean value?Check the Item (delete ee and wk 5 to 9 of age)
Author response:The average is from statistics described in MM section.We have deleted ee. “wk 5 to 9 of age” is a section of the form, Same as “wk 9 to 21 of age”

This manuscript is a resubmission of an earlier submission. The following is a list of the peer review reports and author responses from that submission.